# The Effect of Biochar Addition on the Biogas Production Kinetics from the Anaerobic Digestion of Brewers' Spent Grain

**Marta Dudek [1], Kacper Świechowski [1], Piotr Manczarski [2], Jacek A. Koziel [3] and Andrzej Białowiec [1,*]**

[1] Faculty of Life Sciences and Technology, Institute of Agricultural Engineering, 37/41 Chełmońskiego Str., Wrocław University of Environmental and Life Sciences, 51-630 Wrocław, Poland; martadudek22@gmail.com (M.D.); kacper.swiechowski@upwr.edu.pl (K.Ś.)

[2] Department of Environmental Engineering, Faculty of Building Services, Warsaw University of Technology, Hydro and Environmental Engineering, 00-653 Warsaw, Poland; piotr.manczarski@pw.edu.pl

[3] Department of Agricultural and Biosystems Engineering, Iowa State University, Ames, IA 50011, USA; koziel@iastate.edu

\* Correspondence: andrzej.bialowiec@upwr.edu.pl; Tel.: +48-71-320-5973

**Abstract:** Biochar (BC) addition is a novel and promising method for biogas yield increase. Brewer's spent grain (BSG) is an abundant organic waste with a large potential for biogas production. In this research, for the first time, we test the feasibility of increasing biogas yield and rate from BSG digestion by adding BC, which was produced from BSG via torrefaction (low-temperature pyrolysis). Furthermore, we explore the digestion of BSG with the presence BCs produced from BSG via torrefaction (low-temperature pyrolysis). The proposed approach creates two alternative waste-to-energy and waste-to-carbon type utilization pathways for BSG: (1) digestion of BSG waste to produce biogas and (2) torrefaction of BSG to produce BC used for digestion. Torrefaction extended the short utility lifetime of BSG waste turned into BC. BSG was digested in the presence of BC with BC to BSG + BC weight ratio from 0 to 50%. The study was conducted during 21 days under mesophilic conditions in $n = 3$ trials. The content of dry mass 17.6% in all variants was constant. The kinetics results for pure BSG (0% BC) were: reaction rate constant ($k$) 1.535 d$^{-1}$, maximum production of biogas ($B_0$) 92.3 dm$^3$·kg$^{-1}_{d.o.m.}$ (d.o.m. = dry organic matter), and biogas production rate ($r$), 103.1 dm$^3$·kg$^{-1}_{d.o.m.}$·d$^{-1}$. his preliminary research showed that the highest ($p < 0.05$) $r$, 227 dm$^3$·kg$^{-1}_{d.o.m.}$·d$^{-1}$ was due to the 5% BC addition. This production rate was significantly higher ($p < 0.05$) compared with all other treatments (0, 1, 3, 8, 10, 20, 30, and 50% BC dose). Due to the high variability observed between replicates, no significant differences could be detected between all the assays amended with BC and the variant 0% BC. However, a significant decrease of $B_0$ from 85.1 to 61.0 dm$^3$·kg$^{-1}_{d.o.m.}$ in variants with the high biochar addition (20–50% BC) was observed in relation to 5% BC (122 dm$^3$·kg$^{-1}_{d.o.m.}$), suggesting that BC overdose inhibits biogas production from the BSG + BC mixture. The reaction rate constant ($k$) was not improved by BC, and the addition of 10% and 20% BC even decreased k relatively to the 0% variant. A significant decrease of $k$ was also observed for the doses of 10%, 20%, and 30% when compared with the 5% BC (1.89 d$^{-1}$) assays.

**Keywords:** waste to carbon; waste to energy; biogas; methane; kinetics; torrefaction; biochar; brewer's spent grain; anaerobic digestion; organic waste; waste management

## 1. Introduction

The European Union (EU) produces $956 \times 10^6$ Mg·year$^{-1}$ of the dry biomass. Organic waste generated by biomass utilization has great potential as a feedstock to the 'waste to energy' and 'waste to carbon' circular economy concepts. To date, approximately 46% of biomass is: (a) incorporated back to the soil to maintain organic content, (b) used as animal bedding, and (c) used for energy production [1]. The remaining 54% is used for (d) food production and (e) for animal feeding. Organic waste can be treated to reduce its negative environmental impact and/or to produce valuable products, such as organic fertilizers and biogas for energy production.

The production of biogas based on organic waste is considered as an environmentally friendly technology resulting in the reduction of waste mass, and transformation of organic matter into useful renewable energy, which in turn reduces greenhouse gas emissions [2]. Microorganisms play a vital role in enzyme production that digests biodegradable components of waste (e.g., cellulose, lignin, starch, and complex polysaccharides, proteins, and fats) into simple nutrients—sugars, amino acids, and fatty acids. As the microbes grow and multiply, a large fraction of waste components is converted into heat, metabolic gases, and water. Methane (a major component of biogas) is generated mainly under anaerobic conditions [3,4]. Many factors affect the methanogenesis course and its efficiency. Dobre et al. [5] mentioned six main factors influencing the biogas production: temperature, retention time, pressure of the digester, pH, volatile fatty acids content, and slurry composition. The slurry quality and rate of fermentation depend on substrate-bulk interactions [6]. Panico et al. [6] showed that chemical composition and texture of substrates plays an important role in the anaerobic digestion process. Substrates with a porous texture containing mostly simple carbohydrates are available for microorganism immediately. An example of such type of feedstock may be brewer's spent grain (BSG).

Brewer's spent grain has a large potential for biogas production. The total biogas production potential of brewer's waste streams in the EU is estimated at $12.6$–$39.7 \times 10^9$ MJ per year [7]. Beer brewing is one of the dominating branches of the food production industry. In Poland, waste from beer brewing and ethanol (distilled dry grains, DDGS) represent 33% of all food industry waste [8]. Poland is the third beer producer in the EU, which represents over 10% of the total amount of EU beer produced [9]. BSG represents about 85% of all waste produced during beer production [10]. Approximately 0.2 kg of wet BSG is generated per L of beer produced [11]. The BSG production is $\sim 3.4 \times 10^6$ Mg and $4.5 \times 10^6$ Mg in the EU and U.S., respectively [12].

To date, a large load of biodegradable and wet BSG needs to be processed in a relatively short time, 7–10 d [13]. Waste from breweries is utilized mainly as cattle feed and compost production [14]. BSG is a good medium for microorganisms due to the high sugar content (~50% of the dry mass.) and thus can produce organic acids, ethanol, glycerol, and butanol [1]. However, regardless of the brewery size, the alternative uses of BSG are rare, i.e., 5–10% for rural and urban breweries, respectively [14]. It is only in recent years that BSG is considered as a substrate for biogas production due to a reduction in revenues from cattle feed applications, an increase in electric energy prices, and incentives for renewable energy [15].

Biochar addition was proposed to increase biogas production efficiency produced from the organic fraction of municipal solid waste (OFMSW) [16]. Biochar is a solid product of the pyrolysis/torrefaction process. Pyrolysis is a process of thermochemical decomposition of biomass. It takes place in a non-oxidative condition in the temperature from 200–1000 °C. The processing time depends on the technology and expected properties of resulting biochars [17]. The temperature and heating rate of the process has a significant influence on the quantity and quality of the biochar. The increase in temperature leads to a reduction in the content of hydrogen and oxygen, so the percent content of coal increases. At higher temperatures, the volatilization is stronger, and therefore, the biochar has a higher porosity. The volatilization is also influenced by the heating rate. The pore surface and pore volume are greater for biochar produced at a higher heating rate. An increase in the processing temperature also causes a reduction in mass yield up to 15% [18]. While the torrefaction mass yield is up to 70% [19]. Biochar can be used in a variety of applications, such as adsorbents, catalysts, and soil amendments.

Recent trials show potential in using biochar for mitigation of ammonia emissions from livestock waste while increasing methane [20]. Biochar can also be produced from municipal waste to create a high-energy alternative fuel (carbonized refuse-derived fuel, CRDF) via torrefaction (low-temperature pyrolysis) [21]. Biochar properties can be modified while adjusting corresponding process parameters to achieve properties desired for an application [22].

The addition of biochar to the fermentation process affects the change in the quantity and quality of produced biogas. Research shows that adding biochar to the methane fermentation process leads to a change in the $CH_4$ content in the gas [23]. For digestion of bio-waste, the addition of biochar 5% and 10% in the dry mass, increased biogas production by 5% and 3%, respectively [19]. The addition of commercial charcoal to the digestion of cow dung increased biogas production by up to 34.7% in semi-continuous fermenters [24]. The addition of 1% of biochar in dry mass increased gas production by 31% during 30 d digestion of cattle manure [25]. Furthermore, for cattle manure, with 1% of BC in the dry mass content, the decrease of concentration of methane in biogas by 8% was observed, while the total net methane production increased to 27% [26]. The influence of biochar on the digestion process also depends on the type and method of biochar production. Higher methane production was obtained for 'hydro-char' (produced from thermophilic wheat straw digestate) than 'pyrolytic char' (made from a mixture of paper sludge and wheat husks) [27]. Hydro-char is formed during the thermochemical decomposition of biomass in the absence of oxygen and in the presence of subcritical, liquid water. This process is also called hydrothermal carbonization. At a temperature of 180–250 °C and a pressure of about 20 bar, the mass yield is about 50–80%. Increasing the temperature of the processing will to reduce the amount of hydro-char in favor of liquid and then gaseous fractions. Under standard conditions for slow pyrolysis at 400 °C, the mass yield is 35% [28].

Torrefaction (low-temperature pyrolysis, ~200 to 300 °C) has shown to produce fuel quality biochars from a flammable fraction of municipal solid waste [29–31] or sewage sludge [32,33]. Torrefaction requires significantly less energy to produce biochar than pyrolysis. Therefore, the application of torrefaction biochar has higher potential than pyrolytic when the same level of increase of biogas yield could be determined. Additionally, when the synergistic effect of torrefaction and biogas production from the same substrate is assumed, the residual heat from biogas utilization units may be reused for covering heat demand for torrefaction. Thus, it would seem valid to explore the possibility of increasing biogas yield from BSG digestion by adding torrefaction biochar. Furthermore, it would seem valid to explore the digestion of BSG in the presence of biochars produced from BSG via torrefaction. Neither of these approaches has been reported before. The proposed approach could potentially create two separate pathways to waste-to-energy and waste-to-carbon utilization of a largely untapped resource, i.e., BSG (Figure 1).

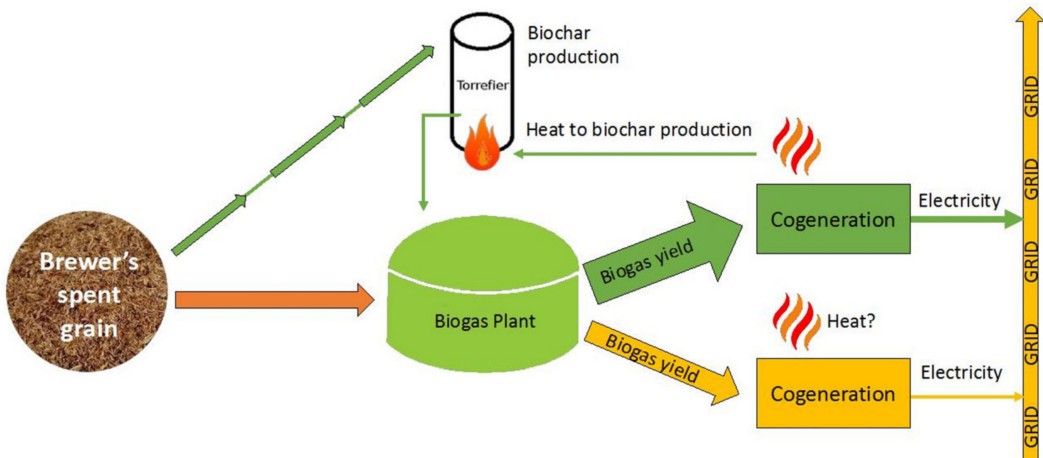

**Figure 1.** The synergistic combination of brewer's spent grain (BSG) torrefaction and anaerobic digestion to increase the biogas production yield.

This work aimed to test whether the torrefaction of part of the BSG into the biochar and its differentiated dose addition to fermentation can improve the efficiency of biogas production. The conducted research was the first step to develop a synergistic strategy of combining the organic waste torrefaction with anaerobic digestion of the same organic waste (in this case, BSG) in the presence of biochar produced from BSG. The study objective was to test the effects of biochar addition on the kinetics of biogas production from anaerobic digestion of BSG. The proposed innovative approach involves the addition of biochar from the BSG torrefaction into the raw BSG to be digested. The proposed approach creates two alternative utilization pathways for BSG utilization: (1) digestion of BSG waste to produce biogas and (2) torrefaction of BSG to produce biochar used for digestion. Torrefaction extends the short utility lifetime of BSG waste and requires less energy input compared with pyrolysis. Therefore, our working hypotheses were (1) biochar addition improves biogas yield, (2) the increase of biochar dose increases biogas production, and (3) the increase of biochar dose increases the biogas production rate.

## 2. Materials and Methods

### 2.1. Used Materials

BSG was used as a substrate. The biochar produced via torrefaction of BSG, and digestate were added. BSG was a by-product of the domestic (from a do-it-yourself kit consisting of barley, yeast, hops, water) beer brewing process. BSG was dried at 105 °C with accordance with PN-EN 12880:2004, by means a laboratory dryer (WAMED, model KBC-65W, Warsaw, Poland). The BSG mass before and after drying was measured with a laboratory balance (Radwag, model As 220. R2, Radom, Poland) with 0.1 g accuracy.

Biochar (BC) was produced from dried BSG. The biochar was made using muffle furnace (SNOL, model 8.1/1100, Utena, Lithuania) via torrefaction, according to a detailed procedure described by Stępień and Białowiec [21]. Parameters of the biochar production process were 300 °C, 60 min and inert gas ($N_2$) flow of 10 L·min$^{-1}$. The liquid digestate (D) from an agricultural biogas plant (Świdnica, Poland), which used sugar beet as a substrate, was used for bioreactors inoculation. The properties of used BSG, BC, and D are presented in Table 1.

**Table 1.** Properties of brewers' spent grain, biochar, and digestate used in the experiment.

| Measured Property | Brewer's Spent Grain (BSG) | Biochar from BSG (BC) | Digestate (D) (Used for Inoculation) |
|---|---|---|---|
| Moisture content in wet BSG, % | 78.7 ± 3.21 | - | - |
| Dry mass content, % | 100 (dried) | 100 (torrefied) | 3.10 ± 0.27 |
| Organic metter content *, % | 96.90 ± 0.80 | 93.30 ± 0.70 | 70.40 ± 0.60 |
| Ash content *, % | 3.10 ± 0.80 | 6.70 ± 0.70 | 29.60 ± 0.60 |
| C content *, % | 50.47 ± 1.56 | 55.90 ± 0.75 | 35.60 ± 0.82 |
| H content *, % | 7.17 ± 0.06 | 5.78 ± 0.17 | 4.30 ± 0.05 |
| N content *, % | 3.63 ± 0.09 | 4.58 ± 0.23 | 4.04 ± 0.23 |
| S content *, % | 0.24 ± 0.01 | 0.23 ± 0.00 | 0.65 ± 0.02 |
| O content *, % | 34.37 ± 0.71 | 24.58 ± 3.28 | 25.81 ± 0.60 |
| Higher heating value*, MJ·kg$^{-1}$ | 20.00 ± 0.30 | 24.90 ± 0.30 | 16.00 ± 0.20 |
| Lower heating value, MJ·kg$^{-1}$ | 17.74 ± 0.25 | 23.6 ± 0.33 | 15.00 ± 0.24 |

* values presented on dry mass basis.

### 2.2. Experimental Setup

The impact of the addition of biochar on the biogas production process was tested using the GB21 method [34,35]. Each test lasted 25 days (including preparing set-up, incubation time of 21 days, materials analysis). The BSG was digested in the presence of BC produced via torrefaction from BSG. A digestate (D) from an agricultural biogas plant was used as a microbiological inoculum to start the fermentation process.

The GB21 ("Gasbildung" in German) indicator is a research method for assessing biogas production [34,35]. The GB21 test provides information on the production of generated biogas

in relation to the mass of the substrate contained in the sample. The test was carried out on the basis of DIN 38414 1985. The process parameters are:

- incubation time = 21 d,
- incubation temperature = 37 °C,
- the total mass of substrates = 35 g.

Due to the dynamic nature of the digestion process, the weight of the analyzed samples was reduced as the digested material (a mixture of BSG, BC, and D) was converted into a gas which exerts measurable differences in headspace pressure.

The GB21 test was performed in $n = 3$ separate trials using the OxiTop®Control AN measuring system (Oxitop Control AN6, Weilheim, Germany) (Figure 2) in all variants, while variant 0% BC was tested four times. The OxiTop®Control AN system is based on sealed vessels with manometric heads calibrated to measure the pressure difference as a result of biogas production. The continuous pressure measurements record was downloaded by the IR interface every one to three days depending on the digestion phase using the OxiTop OC 110 controller (Weilheim, Germany). Side connectors allowed for the initial introduction of the inert gas ($CO_2$) into the vessel and also to release pressure buildup. The constant mesophilic temperature of 37 °C inside the vessels was maintained using an incubation chamber (Pollab, model 140/40, Wilkowice, Poland).

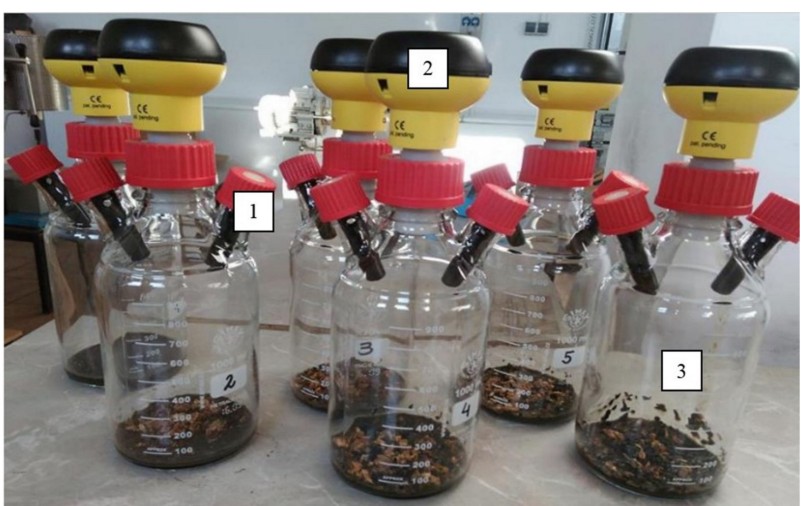

**Figure 2.** Biogas production from a mixture of BSG, biochar produced via torrefaction of BSG, and digestate inside an OxiTop Control AN6 system. Replicated trials were conducted for 21 days at 37 °C. **1**—side connectors, **2**—manometric head with a pressure sensor and data storage, **3**—sealed glass vessel.

### 2.3. Experimental Design

The study was based on a comparison of the biogas production kinetic parameters between individual variants. The mixtures (BSG + BC + D) were prepared according to the experimental matrix (Table 2) and placed in test vessels OxiTop Control AN6. The samples in vessels were then flushed with $CO_2$ inert gas to create anaerobic conditions in each vessel. Digestate (D) without BSG or BC was used as a control variant, considered to account for the inoculum influence in biogas production itself. Dry mass content in D variant was 3.06% (Table 1). Thus, the biogas production in relation to kg dry organic matter in D was subtracted (point by point) from results obtained from other variants. Other variants were prepared with the mixture of D (85% wet mass) and BSG + BC (15% dry mass, BSG and BC were after drying and torrefaction, respectively). The initial mass fraction of BC in the BSG + BC mixture increased from 0 (variant S0) to 50% (variant S50) (Table 2), but the share of BSG + BC in the mixture BSG + BC + D was constant 15%, with the inoculum (D) to mixture (BSG + BC) mass (g/g)

ratio 29.75/5.25 = 5.67 (Table 2). The initial dry matter content in each variant was 17.6% to achieve conditions of wet fermentation.

**Table 2.** Research variants, initial weight fraction of individual components (0% BC (S0) to 50% BC (S50) addition), g.

| Components | Variant (Mass, g) | | | | | | | | | |
|---|---|---|---|---|---|---|---|---|---|---|
| | D (Control) | S0 | S1 | S3 | S5 | S8 | S10 | S20 | S30 | S50 |
| D * | 30.0000 | 29.7500 | 29.7500 | 29.7500 | 29.7500 | 29.7500 | 29.7500 | 29.7500 | 29.7500 | 29.7500 |
| BSG ** | 0.0000 | 5.2500 | 5.1975 | 5.0925 | 4.9875 | 4.8300 | 4.72500 | 4.2000 | 3.6800 | 2.6250 |
| BC ** | 0.0000 | 0.0000 | 0.0525 | 0.1575 | 0.2625 | 0.4200 | 0.52500 | 1.0500 | 1.5700 | 2.6250 |

\* wet mass of digestate was given, ** dry mass of BSG and BC was given, as these materials were dried and torrefied, respectively.

## 2.4. Standard Analytical Methods

The BSG, BC, and D components were characterized by the following standard methods:

- Organic matter was determined in accordance with PN-EN 15169:2011 using the muffle furnace (SNOL, model 8.1/1100, Utena, Lithuania),
- Ash content in accordance with the PN-G-04516:1998 standard using the same muffle furnace and Radwag PS 3500.R2 analytical balance,
- Elementary composition of C, H, N, O, and S, using Perkin Elmer 2400 Series CHNS/O with Radwag, MYA 2.4 Y analyzer in accordance with PN-EN 15104: 2011
- Lower heating value and higher heating value by means of the IKA C2000 Basic calorimeter in accordance with the PN-Z-15008-04:1993 standard.

The properties of the organic matter content in the initial mixture and residual material were measured according to procedure PN-EN 15169:2011, with using the muffle furnace (SNOL, model 8.1/1100, Utena, Lithuania).

## 2.5. Data analysis—Assessment of Biogas Production Potential

The conducted experiment allowed to determine the potential of biogas production. The data read from the OxiTop OC 110 controller presented pressure changes over time every 120 s, taking place inside the vessel with a capacity of $1.1 \times 10^{-3}$ m$^3$. The data was used to determine the potential of GB21. The GB21 index is defined as dm$^3$ of biogas per kg of dry organic matter [34,35]. Data from the OxiTop OC 110 controller was processed using software Microsoft Office Excel. Individual variant's pressure changes as a function of time were determined from raw data. The number of moles of gas produced in the vessel was determined using the Clapeyron equation (ideal gas law) (Equation (1)).

$$n_g = \frac{p \cdot V_v}{R \cdot T} \tag{1}$$

where:

$n_g$—the number of moles of gas in the vessel, mol,
$p$—accumulated gas pressures in the vessel, Pa,
$V_v$—the volume of the vessel, m$^3$
$R$—universal gas constant ($R = 8.314$ J·mol$^{-1}$·K$^{-1}$), J·mol$^{-1}$·K$^{-1}$
$T$—incubation temperature, K.

The volume of generated biogas was determined by Avogadro's law with Equation (2).

$$V_g = V_{mol} \cdot n_g \tag{2}$$

where:

$V_g$—the volume of biogas produced, $m^3$,
$V_{mol}$—the molar volume of gas under normal conditions ($V_{mol} = 0.0224 \ m^3$), $m^3$,

The value of the GB21 index for individual volumes of gas accumulated over time was determined based on Equation (3).

$$GB21 = \frac{V_g}{m_{d.o.m.}} \tag{3}$$

where:

$GB21$—the amount of gas produced from 1 kg of dry organic matter (BSG), $dm^3 \cdot kg^{-1}{}_{d.o.m.}$,
$m_{d.o.m.}$—dry organic mass in substrates, kg.

The value of the reaction rate constant ($k$) was determined by the non-linear estimation of the experimental data (GB21) to the first order cumulative equation (Equation (4)) according to [36–40]. Estimation of the best-fit parameters was carried out using the Statistica 12 software (StatSoft Polska, Krakow, Poland).

$$B_t = B_0 \cdot \left(1 - e^{-k \cdot t}\right) \tag{4}$$

where:

$B_t$—the amount of biogas obtained from the substrate (BSG) after process time t, $dm^3 \cdot kg^{-1}{}_{d.o.m}$,
$B_0$—maximum production of biogas from the substrate, $dm^3 \cdot kg^{-1}{}_{d.o.m}$,
$k$—reaction constant rate, $d^{-1}$,
$t$—time, d.

The biogas production rate ($r$), $dm^3 \cdot kg^{-1}{}_{d.o.m.} \ d^{-1}$ has been determined based on (Equation (5)).

$$r = k \cdot B_0 \tag{5}$$

where:

$r$—biogas production rate, $dm^3 \cdot kg^{-1}{}_{d.o.m.} \ d^{-1}$,

The detail ANOVA evaluation of differences between mean values in relation to biochar content in the feedstock (%) was performed with application of post-hoc lowest significant difference (LSD) test, at the $p < 0.05$ significance level using Statistica 12 software (StatSoft, Inc., TIBCO Software Inc., Palo Alto, CA, USA).

## 3. Results

### 3.1. Assessment of Biogas Production Potential

The course of cumulative biogas production has been presented in Appendix A (Figures A1–A9). The lag-phase has not been observed in all repetitions of tested variants. The biochar addition had an effect on biogas production, which depended on BC dose. This statement should be clarified by stating that used vessels were considered as the "black box" system while we measured the biogas accumulation as an effect of biological activity. The mean $B_0$ (maximum production of biogas) value ranged from 61.0 to 122.0 $dm^3 \cdot kg^{-1}{}_{d.o.m.}$ Biogas production for BSG only variant (0% BC) resulted in 92.3 $dm^3 \cdot kg^{-1}{}_{d.o.m.}$ The 5% BC addition resulted in the highest $B_0$ (122.0 $dm^3 \cdot kg^{-1}{}_{d.o.m.}$) which was significantly higher ($p < 0.05$) than $B_0$ in variants with the lowest biogas production (3% and 50% BC) but was not significantly ($p > 0.05$) higher than in the 0% BC variant. The lack of observations of the significant influence between these variants could be a result of the relatively high variability of the results, which was visualized by a high range of standard deviations (Figure 3). Further research with more repetitions should continue.

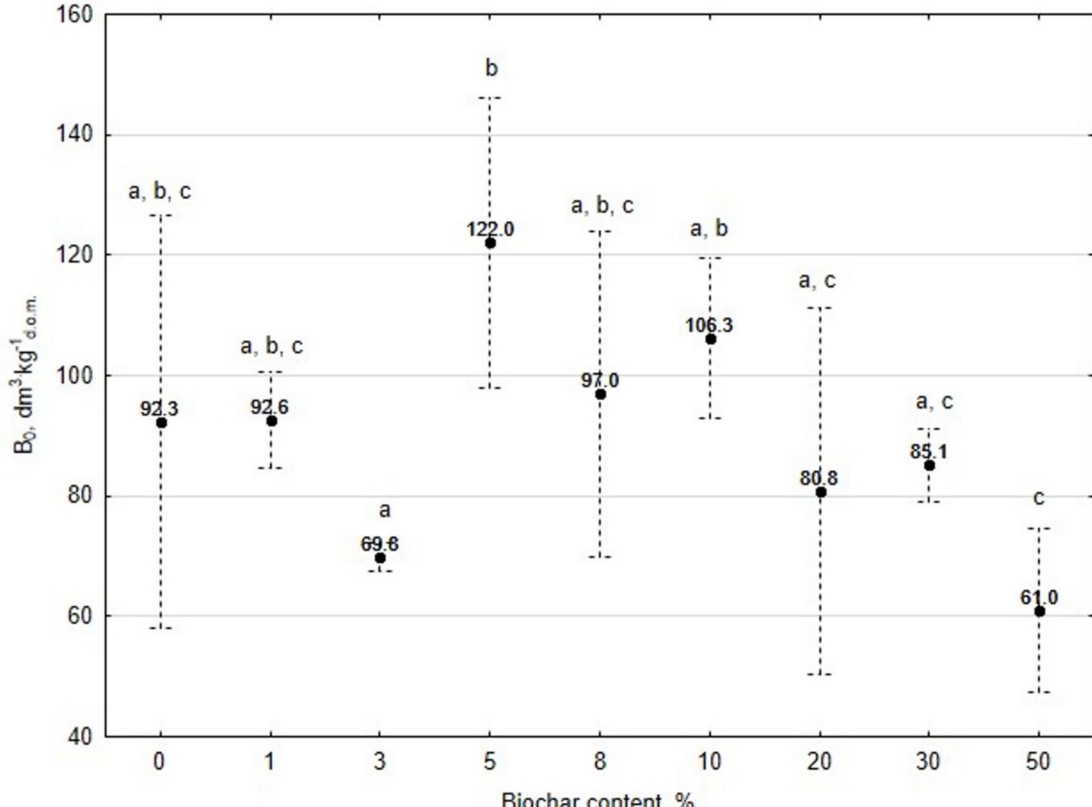

**Figure 3.** The effect of biochar (BC) mass content (%) addition to brewer's spent grain (BSG) digestion on maximum production of biogas from the substrate ($B_0$, $dm^3 \cdot kg^{-1}_{d.o.m.}$). Letters denote significant differences ($p < 0.05$) between mean values.

The increase of BC content above 10% did not significantly ($p < 0.05$) change $B_0$ to values lower than in a variant without BC addition. The increase of BC content to values in the range of 20–50% significantly ($p < 0.05$) reduced $B_0$ in comparison to variant 5% BC. The decreasing trend of $B_0$ in variants over 8% BC in comparison to 5% (which was statistically significant ($p < 0.05$) in the range of 20 to 50% BC), could be caused by the occurrence of Maillard reaction during biochar generation.

Maillard reaction takes place in the thermal process between amino acids and sugars [41] resulting in the production of melanoidins—substance hardly degradable anaerobically [41–43]. BSG is rich in carbohydrates (~50% of the dry mass [44]) and proteins (19–30% of the dry mass [45]). These compounds could react and transform into organic compounds not available for microorganisms. Such an explanation may be derived from the experiment of Ariunbaatar et al. [41] who tested thermally treated (80–150 °C) food waste biogas production potential. The thermal pretreatment resulted in an increase in cumulative biogas production. The biggest increase was obtained for 80 °C, over 22% in methane production in comparison to not treated food waste. However, the rise of treatment temperatures caused a decrease in methane production in comparison to 80 °C. Bougrier et al. [46] proposed that the thermal pretreatment could also cause a reaction between the soluble carbohydrates and soluble proteins, forming Amadori-like compounds [47]. These Amadori compounds are the by-products of melanoidins [43,47–49], and the formation of such compounds might have also yielded a lower biogas production [41].

The role of the Maillard reaction in this experiment cannot be fully determined. The BSG used in the presented work was torrefied at 300 °C. BSG has a high content of carbohydrates and proteins [44,45]. The thermal degradation of amino acids (substances that build proteins) takes place between 185 and 280 °C [50]. According to [51] thermal degradation of protein and carbohydrates takes place in 209–309 °C and 164–497 °C, respectively. Therefore, theoretically, Maillard reaction could take place during the

BC generation, but mostly during the increase of temperature from the initial room temperature to the setpoint (300 °C). It is likely that after the temperature had reached 300 °C and remained there for 60 min, all of the carbohydrates, proteins, and probably melanoidins and Amadori-like compounds were thermally degraded. In addition, another factor or parameter of BC could negatively affect the biogas production process. Such factor could be the presence of pollutants, and therefore, BC toxicity [52]. The idea of the synergy of biochar application for biogas production enhancement requires further fundamental studies on reactions during organic waste torrefaction, the final composition of biochar, the content of organic, and inorganic pollutants, and toxicity and bioavailability of these compounds. Further research is warranted, with a wide range of temperatures and torrefaction times used for BC production and more repetitions.

The reaction rate constant ($k$) increased slightly (albeit without statistical significance, $p < 0.05$) from 0 to 5% BC addition. Estimated $k$ values were relatively high, which could be explained by the higher bioavailability of more biodegradable components of the feedstock. Further increase of BC content decreased the biogas production constant rate, especially in the 10 and 20% BC content variants which were significantly lower ($p < 0.05$) compared with 0% and 5% BC (Figure 4).

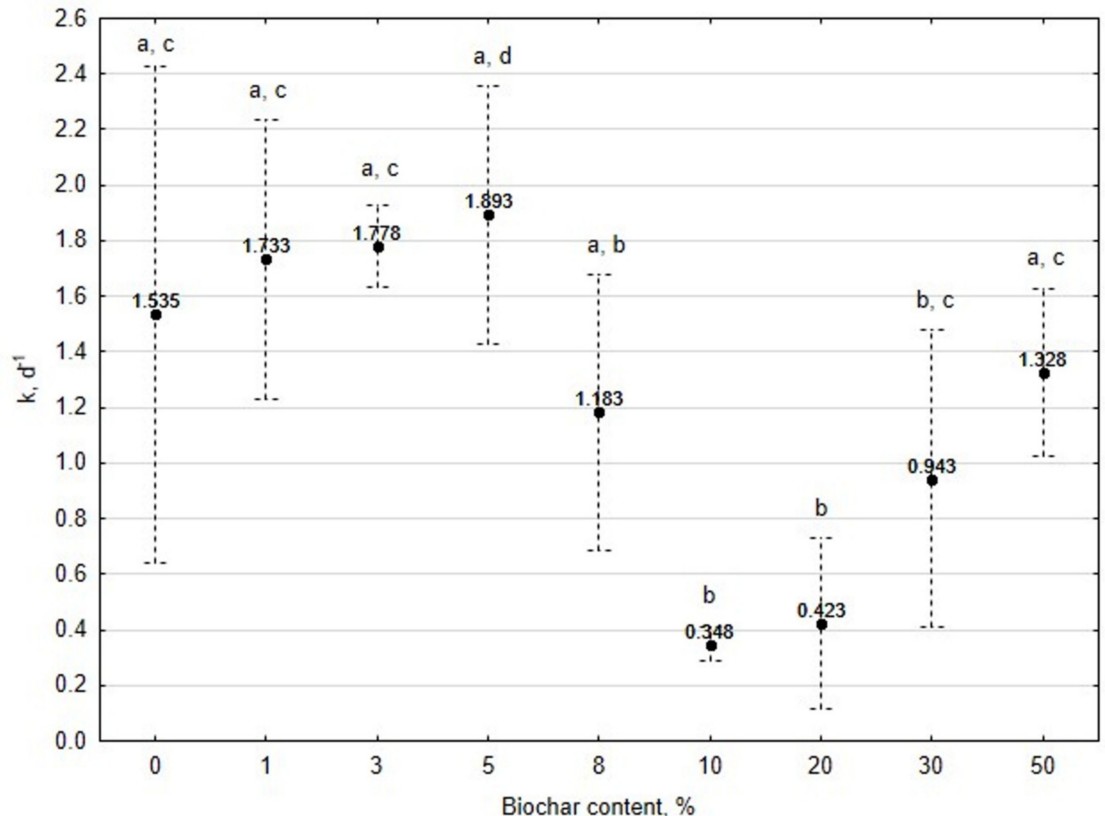

**Figure 4.** The influence of biochar (BC) mass content (%) addition to brewer's spent grain (BSG) digestion on reaction rate constant ($k$), d$^{-1}$. Letters show significant differences ($p < 0.05$) between mean values.

The estimation of biogas production rate ($r$) showed that the statistically significant ($p < 0.05$) most effective variant was 5% BC addition to BSG. All other variants had a lower $r$ and were statistically different (Figure 5). Variants with 10% and 20% of BC had significantly lower $r$ compared with 0% BC.

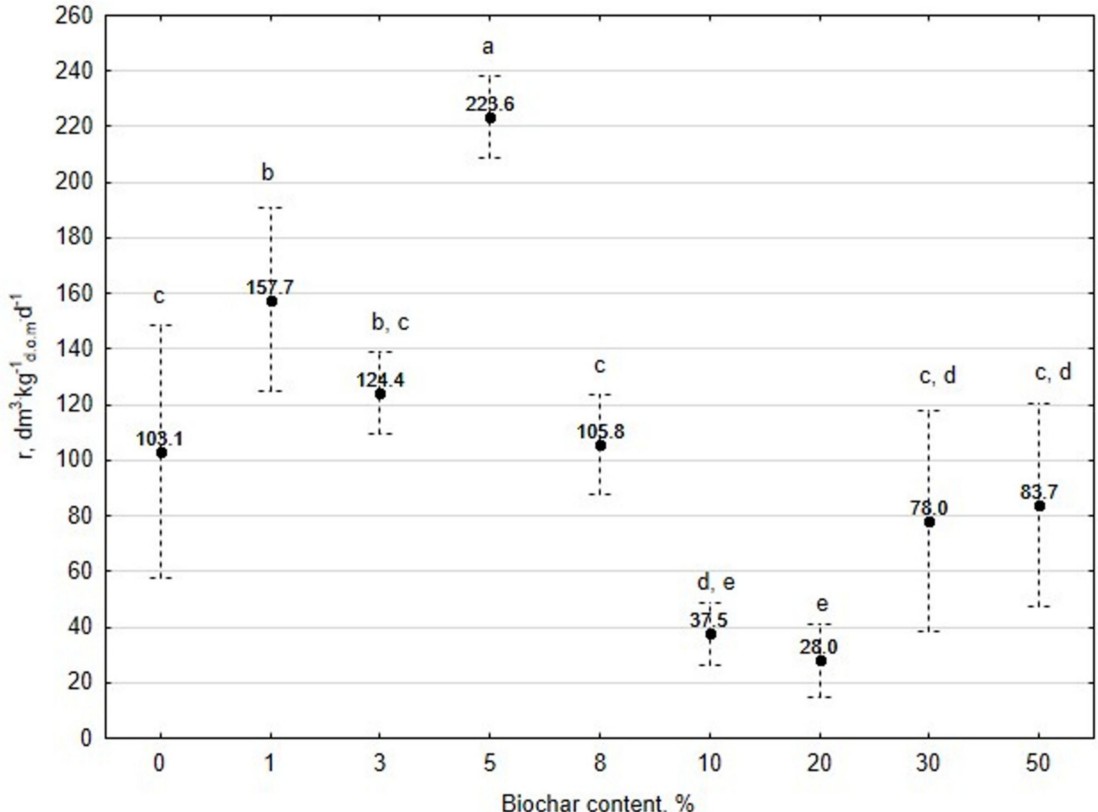

**Figure 5.** The influence of biochar (BC) mass content (%) addition to brewer's spent grain (BSG) digestion and biogas production rate ($r$), $dm^3 \cdot kg^{-1}_{d.o.m.} \cdot d^{-1}$. Letters show significant differences ($p < 0.05$) between mean values.

Based on these observations (Figures 2–4), high biochar addition (20–50% BC) decreased the biogas production.

The total C/N ratio in all tested variants was almost the same (i.e., 11 ± 0.25). Theoretically, microorganisms should have had similar nutrition conditions. Moreover, it was shown [53] that for anaerobic digestion of municipal solid waste, the optimum C/N ratio in solid substance should be ~10. Thus, all variants could be characterized by a sufficient nutrition level, which excludes problems with lack of biogens leading to methanogenesis deviation.

It can also be hypothesized that at a certain (low) BC dose, the biochar enhances waste digestion (fermentation) and biogas production processes, which was observed especially for 5% BC. Work [17] showed that 5% BC addition gave better (~2%) biogas yield than 10% BC addition to bio-waste anaerobic digestion. The positive effect, up to 69%, on the increase of biogas yield, was shown by Pan et al. [54] who tested different types of biochars produced from wheat straw, discarded fruitwood and chicken manure under temperatures of 350, 450 and 550 °C. This also confirms that there appears to be a limit beyond which the properties of the biochar contribute to inhibiting the fermentation process or biogas net production.

In a future study, the biodegradability of the biochar should be tested. This is important to address because in this research the increase of the biochar dose was accompanied by a decrease of the BSG dose (Table 2). Thus, the results presented could be potentially affected by biodegradability of biochar, as during torrefaction up to ~3% of biomass could be converted to acetic acid and other organic compounds [55]. In this research, we assumed that biochar is not biodegradable. This assumption needs to be tested because it is possible that some unknown fraction of biochar is digested, and therefore, increasing the biogas production. Further investigation of the mechanism of biochar properties influence on methane fermentation and net biogas yield is recommended.

Based on this preliminary experiment, it is recommended to execute further research on the influence of biochar addition on biogas production in the range below 10%, with the interval of 1%, as higher doses of BC from torrefed BSG do not appear to improve biogas yield. Investigating the mechanism of biochar addition on the biogas fermentation process, influence on microbial activity, and biogas net yield due to chemical–physical interactions with biogas components is also warranted. Further investigation could also consider the application of other types of biochar for digestion of BSG and other types of organic waste, as it opens new approaches to biogas yield increase and new possibilities of co-utilization of biochar and digestate in agriculture (Figure 6).

### 3.2. Simplified Energy Balance Comparing Digestion with and without Biochar from BSG

We propose a simplified energy balance evaluation for comparison of two scenarios. We used the preliminary research on the proof-of-the-concept of a synergistic combination of anaerobic digestion of BSG in the presence of BC (obtained from torrefaction of BSG) for anaerobic digestion enhancement.

1. Scenario (1): anaerobic digestion of BSG in the presence of BC (obtained with torrefaction of BSG);
2. Scenario (2): anaerobic digestion of BSG.

First, the heat demand for BSG drying and torrefaction was estimated according to the following assumptions:

- BSG moisture content = 78.7% (Table 1); 1 ton of BSG contains 787 kg of water, which must be removed.
- The specific heat capacity of water = 4.2 kJ $(kg\,°C)^{-1}$;
- The specific heat of BSG = 2.0 kJ $(kg\,°C)^{-1}$; the specific heat capacity of BSG has not been reported yet. Thus, we used the specific heat capacity of barley ~1.2 kJ $(kg\,°C)^{-1}$ [56] and increased it to be conservative;
- The temperature gradient between room temperature of 20 °C and the water boiling point 100 °C is 80 °C;
- The heat of water evaporation is 2257 kJ $kg^{-1}$.

Based on these assumptions, the energy demand for water heating ($E_w$), solid fraction heating ($E_s$) and water evaporation ($E_v$) were calculated as follows:

$$E_w = 80 \cdot 4.2 \cdot 787 = 264,432 \text{ kJ} \tag{6}$$

$$E_s = 80 \cdot 2.0 \cdot (1000 - 878) = 34,080 \text{ kJ} \tag{7}$$

$$E_v = 2257 \cdot 787 = 1,776,259 \text{ kJ} \tag{8}$$

Thus, the total emery demand for drying ($E_d$) is equal to 2,074,771 kJ.

Next, we assumed, that a typical torrefaction mass yield is 75% [29]. We also assumed that torrefaction is combined with drying, and the dried solid fraction does not cool significantly. The initial temperature before torrefaction is 90 °C. The temperature of torrefaction is 300 °C, as was used in the present experiment to produce BC. Therefore, the temperature gradient for torrefaction is 210 °C. Thus, assuming the same specific heat capacity of the solid fraction to be 2.0 kJ $(kg\,°C)^{-1}$, the energy demand for torrefaction ($E_t$) may be calculated as follows:

$$E_t = 210 \cdot 2.0 \cdot (1000 - 787) = 89,460 \text{ kJ} \tag{9}$$

Assuming the torrefaction mass yield 75%, the BC production should be ~160 kg from one ton of wet BSG. Total energy demand for BSG drying and torrefaction is 2,164,231 kJ per ton of wet BSG or per 160 kg of produced biochar. For the production of 1 kg of biochar 13,526.4 kJ of energy must be used.

Estimated energy demand has been used for comparing the energy balance of the two scenarios.

3.2.1. Scenario 1: Anaerobic Digestion of BSG in the Presence of BC (Obtained with Torrefaction of BSG)

The most efficient variant 5% of BC mixed with BSG is assumed. Initial assumptions:

- Assuming the mass of used BC as 1 kg, the mass of BSG should be 19 kg;
- The content of dry organic matter in BSG is 96.9% (Table 1);
- The $B_0$ from BSG in 5% BC variant was 122 dm$^{-3}$.kg$^{-1}$d.o.m (Figure 3).

Biogas production yield ($B_1$) from BSG is:

$$B_1 = 19 \cdot 0.969 \cdot 122/1000 = 2.246 \text{ m}^3 \tag{10}$$

Assuming that 1.33 kg of dried BSG must be used for the production of 1 kg of BC (torrefaction mass yield 75%), the biogas yield is, therefore, reduced by the biogas ($B\_$) that was not produced from 0.33 kg of dried BSG required for BC production. The biogas that was not produced biogas may be estimated as follows:

$$B_- = 0.33 \cdot 0.969 \cdot \frac{122}{1000} = 0.03 \text{ m}^3 \tag{11}$$

Therefore, the net biogas yield is 2.216 m$^3$. Assuming that biogas lower calorific value is 20,000 kJ m$^{-3}$, the available energy in biogas ($E_b$) is:

$$E_b = 2.216 \cdot 20,000 = 44,320 \text{ kJ} \tag{12}$$

For biogas utilization, the CHP unit with electricity and heat efficiency 38% and 40%, respectively, may be used. Thus, the electricity production ($E_e$) is:

$$E_e = 44,320 \cdot 0.38 = 16,842 \text{ kJ} \tag{13}$$

The heat recovery ($E_h$) is:

$$E_e = 44,320 \cdot 0.4 = 17,728 \text{ kJ} \tag{14}$$

Assuming the energy demand for production of 1 kg of BC ($E_d + E_t$) is 1356.4 kJ, recovered heat ($E_h$) covers all heat demand for drying, torrefaction process required for the production of 1 kg of BC used for digestion with 19 of BSG (5% BC). Additionally, heat reuse rate (generated in CHP) is 76.3%. This heat reuse may be considered as a renewable energy source.

3.2.2. Scenario 2: Only BSG is Anaerobically Digested

Initial assumptions:

- 20 kg of BSG is used for anaerobic digestion.
- The content of dry organic matter in BSG is 96.9% (Table 1)
- The $B_0$ from BSG in variant without BC was 92.3 dm$^{-3}$ kg$^{-1}$d.o.m. (Figure 3).

Biogas production yield ($B_2$) from BSG is:

$$B_2 = 20 \cdot 0.969 \cdot 92.3/1000 = 1.789 \text{ m}^3 \tag{15}$$

Assuming that biogas lower calorific value is 20,000 kJ·m$^{-3}$, the available energy in biogas ($E_b$) is:

$$E_b = 1.789 \cdot 20,000 = 35,780 \text{ kJ} \tag{16}$$

For biogas utilization the CHP unit with electricity and heat efficiency 38% and 40%, respectively, may be used. The electricity production ($E_e$) is:

$$E_e = 35,780 \cdot 0.38 = 13,596 \text{ kJ} \tag{17}$$

which is lower of about 3246 kJ (19%) than in Scenario 1.

The heat recovery ($E_h$) is:

$$E_e = 35,780 \cdot 0.40 = 14,312 \text{ kJ} \tag{18}$$

In this scenario, the additional application of heat should be found.

Presented simplified energy balance shows that the variant with a synergistic combination between anaerobic digestion and torrefaction of the same organic waste, and application of produced biochar for enhancement the anaerobic digestion is a more efficient solution. It could generate more electricity and allows to utilize generated heat, considered as a renewable source of energy. Further research in this concept should be continued with both BSG + BC (from BSG) and with another type of waste (Figure 6).

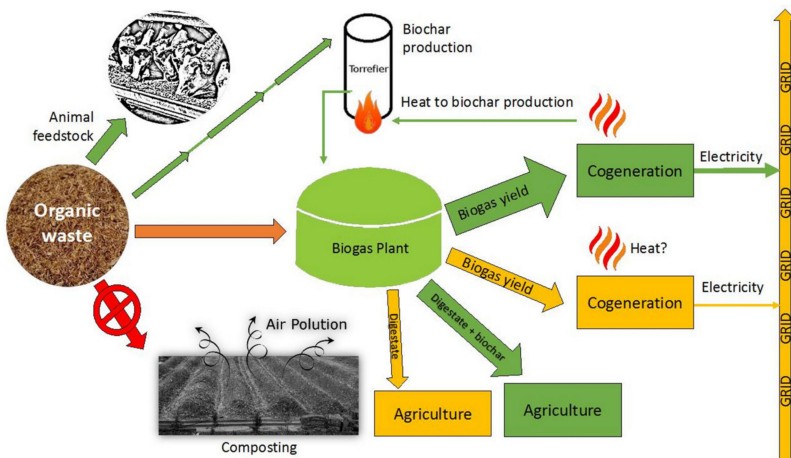

**Figure 6.** The new approach to organic waste utilization based on a synergistic combination of organic waste torrefaction and anaerobic digestion as a potential concept of biogas yield increase, utilization of heat from biogas reuse in cogeneration units, and application new type of organic fertilizer containing the digestate and biochar.

### 3.3. Changes of Mixtures Properties Due to Anaerobic Digestion

The content of organic matter in process residues was 89.7–92.2% (Table 3). The estimated organic matter removal efficiency was in the range between 4.18%, in the case of variant 50% BC, and 5.76%, in the case of variant 10% BC. Results showed that the organic matter removal efficiency is not related to biogas production potential $B_0$, and rate ($r$), as the estimated correlation coefficients were at the level 0.585, and 0.416 respectively. It indicates that the mechanism of biogas net (observed) cumulation could be an effect of biogas biological production (gross), but also other physical, and chemical processes related to biochar activity (e.g., sorption) [20]. This influence requires further study.

**Table 3.** Organic matter content reduction due to anaerobic digestion.

| Parameter | Variant | | | | | | | | |
|---|---|---|---|---|---|---|---|---|---|
| | S0 | S1 | S3 | S5 | S8 | S10 | S20 | S30 | S50 |
| Initial OM * content, % | 96.2 ± 0.75 | 96.1 ± 0.75 | 96.1 ± 0.75 | 96.0 ± 0.74 | 95.9 ± 0.74 | 95.8 ± 0.74 | 95.5 ± 0.74 | 95.1 ± 0.73 | 94.4 ± 0.72 |
| Final OM * content, % | 91.41 ± 2.78 | 92.21 ± 2.39 | 91.00 ± 1.49 | 90.98 ± 0.27 | 90.86 ± 0.46 | 90.60 ± 0.23 | 90.66 ± 0.51 | 90.78 ± 0.27 | 90.64 ± 0.41 |
| Relative change, % | 5.21 | 4.26 | 5.56 | 5.51 | 5.54 | 5.76 | 5.31 | 4.78 | 4.18 |

\* OM—organic matter

## 4. Conclusions

This preliminary research on the effect of the addition of BC produced due to torrefaction of BSG to determining the biogas production due to anaerobic digestion of BSG allowed to conclude that:

- The highest biogas production rate ($r$) resulted due to the 5% BC addition and it was 227 $dm^3 \cdot kg^{-1}_{d.o.m.} \cdot d^{-1}$. This production rate was significantly higher ($p < 0.05$) compared with all other treatments (0, 1, 3, 8, 10, 20, 30, and 50% BC).
- The 5% BC dose resulted in the maximum production of biogas from the substrate ($B_0$) of 122 $dm^3 \cdot kg^{-1}_{d.o.m.}$ but it was not significantly different from 0% BC variant. The significant ($p < 0.05$) inhibition of $B_0$ in comparison to 5% BC was found for variants with 20–50% BC.
- The 5% BC dose resulted in the highest reaction rate constant ($k$) 1.89 $d^{-1}$, but it was not significantly different from 0% BC variant. The significant ($p < 0.05$) decrease of the $k$ was found for variants 10%, 20%, and 30% BC in relation to 5% BC.
- The high biochar addition (20–50% BC) did significantly decrease biogas production in relation to variant 5% BC.

It is recommended to continue the research focused on lower doses of biochar addition from 0 to 10% in a higher number of repetitions to decrease the variability of the results. It is also recommended to investigate the mechanism of biochar addition in the methane fermentation process, influence on microbial activity, and biogas net yield due to chemical-physical interactions with biogas components, including the influence of the Maillard reaction products due to BSG torrefaction. Further investigation could also consider the application of other types of biochar for digestion of BSG. Simplified energy balance modeling indicated that scenario with a synergistic combination of anaerobic digestion of BSG in the presence of BC (obtained from the torrefaction of BSG) and application of produced biochar from BSG for anaerobic digestion enhancement is more efficient than BSG anaerobic digestion itself. This promising concept requires further study.

**Author Contributions:** Conceptualization, A.B. and P.M.; methodology, A.B., M.D. and P.M.; formal analysis, A.B. and J.K.; validation, M.D., K.Ś., A.B. and J.K.; investigation, M.D., K.Ś. and P.M.; resources, M.D. and K.Ś.; data curation, A.B. and J.K.; writing—original draft preparation, M.D. and K.Ś.; writing—review and editing, A.B., P.M. and J.K.; visualization, A.B. and M.D.; supervision, A.B. and J.K.

**Funding:** This research received no external funding.

**Acknowledgments:** Authors would like to thank the Fulbright Foundation for funding the project titled "Research on pollutants emission from Carbonized Refuse Derived Fuel into the environment," completed at the Iowa State University. In addition, this paper preparation was partially supported by the Iowa Agriculture and Home Economics Experiment Station, Ames, IA, USA. Project no. IOW05556 (Future Challenges in Animal Production Systems: Seeking Solutions through Focused Facilitation) sponsored by Hatch Act and State of Iowa funds. Also, the revisions of the manuscript were done thanks to The PROM Programme—International scholarship exchange of Ph.D. candidates and academic staff" co-financed by the European Social Fund under the Knowledge Education Development Operational Programme PPI/PRO/2018/1/00004/U/001.

**Conflicts of Interest:** The authors declare no conflict of interest. The funders had no role in the design of the study; in the collection, analyses, or interpretation of data; in the writing of the manuscript, or in the decision to publish the results.

## Appendix A

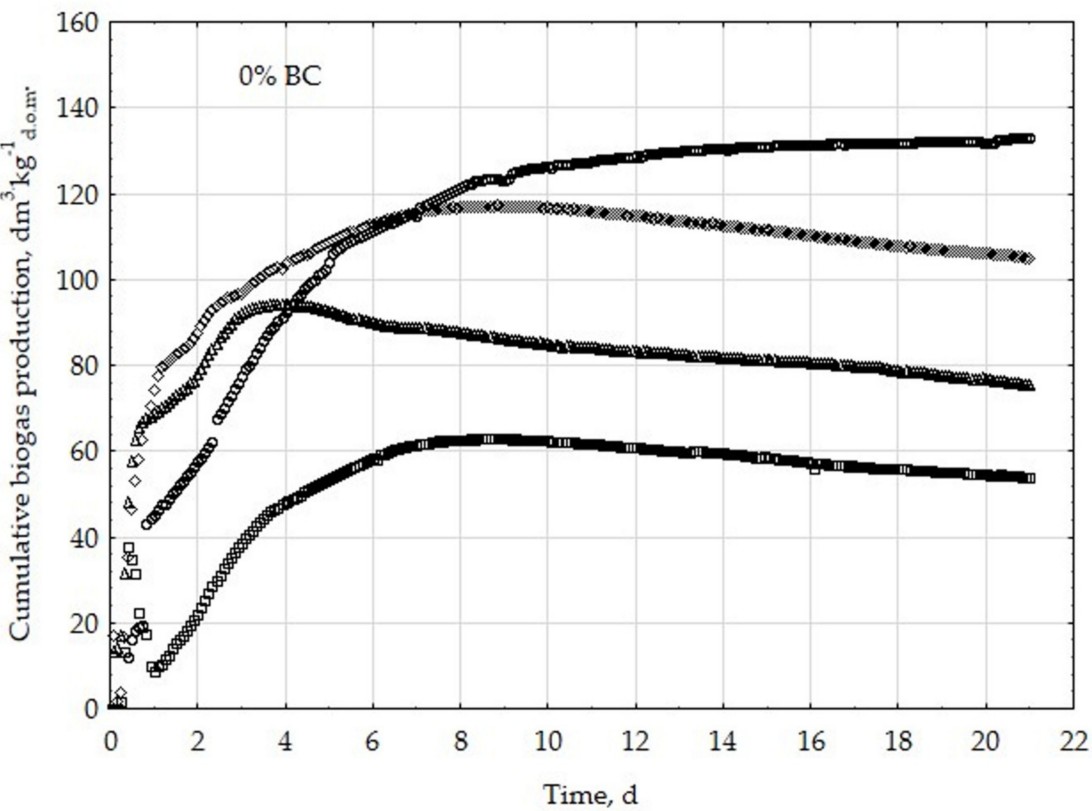

**Figure A1.** Biogas production from BSG: cumulative curves in variant with biochar addition at the level of 0%. The number of repetitions was four.

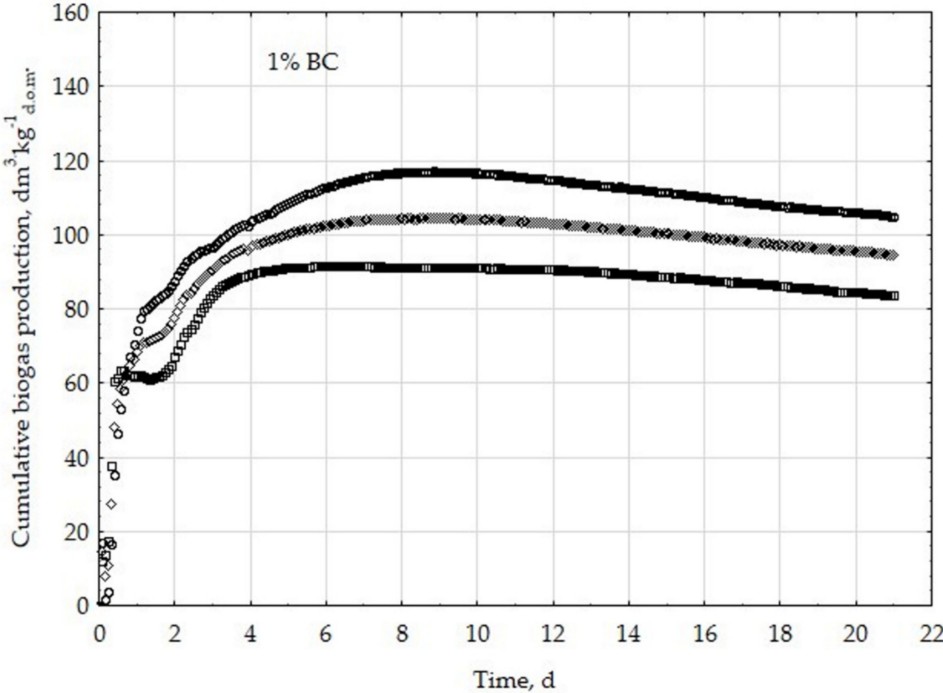

**Figure A2.** Biogas production from BSG: cumulative curves in variant with biochar addition at the level of 1%. The number of repetitions was three.

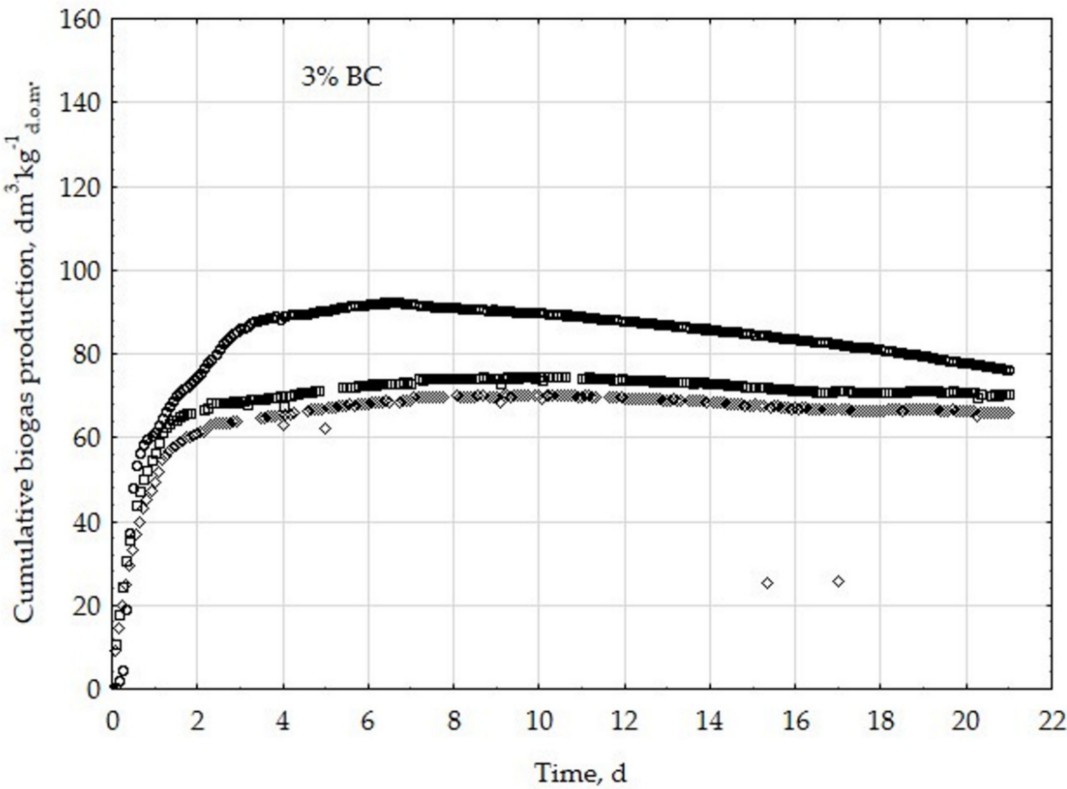

**Figure A3.** Biogas production from BSG: cumulative curves in variant with biochar addition at the level of 3%. The number of repetitions was three.

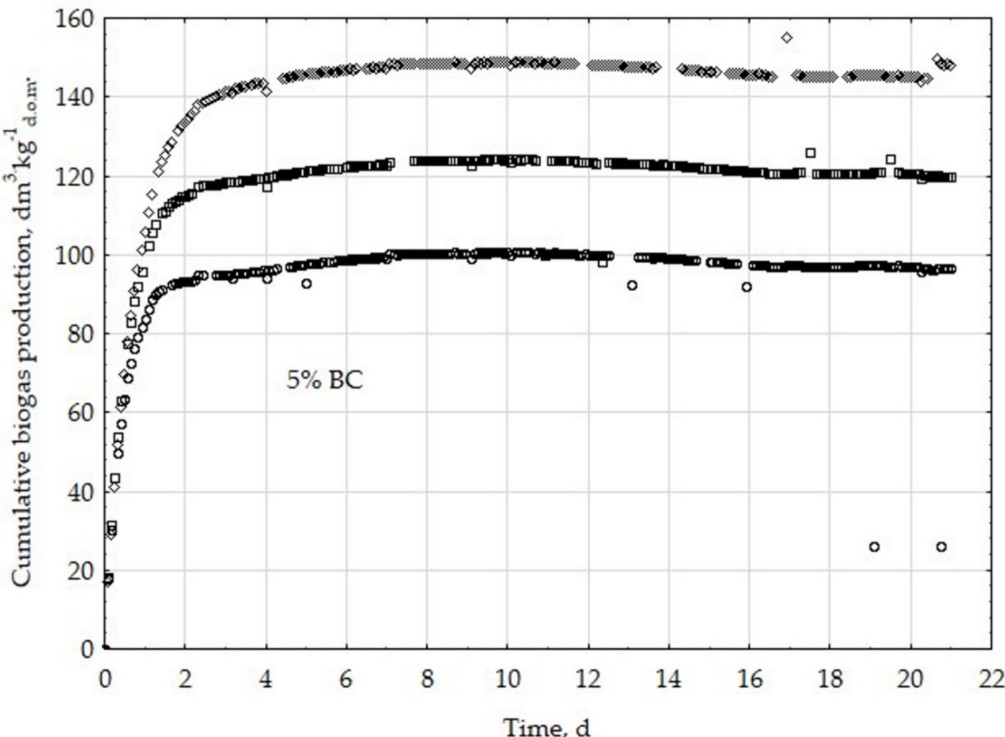

**Figure A4.** Biogas production from BSG: cumulative curves in variant with biochar addition at the level of 5%. The number of repetitions was three.

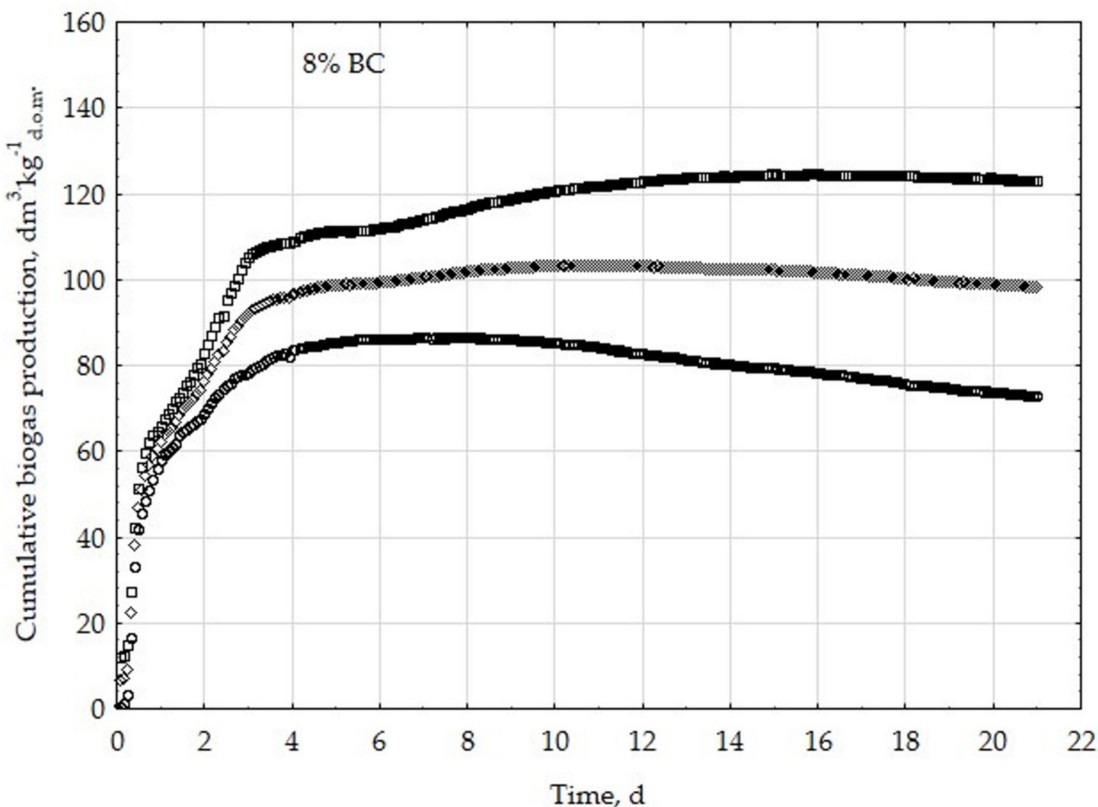

**Figure A5.** Biogas production from BSG: cumulative curves in variant with biochar addition at the level of 8%. The number of repetitions was three.

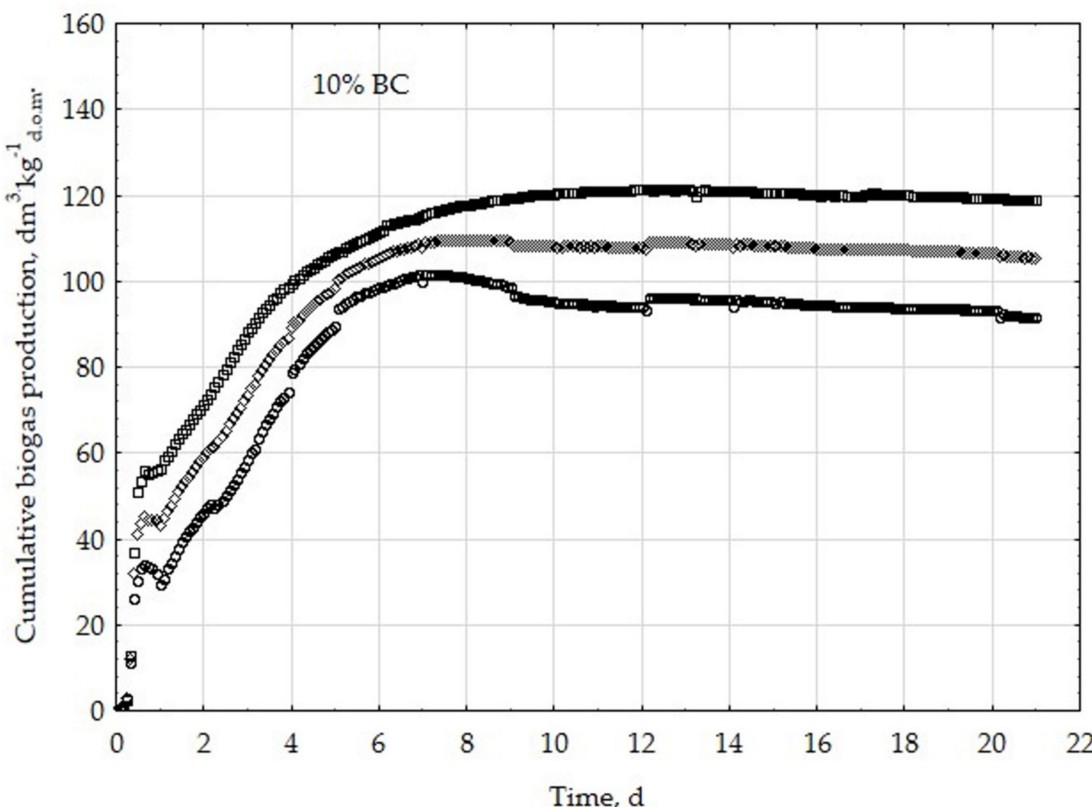

**Figure A6.** Biogas production from BSG: cumulative curves in variant with biochar addition at the level of 10%. The number of repetitions was three.

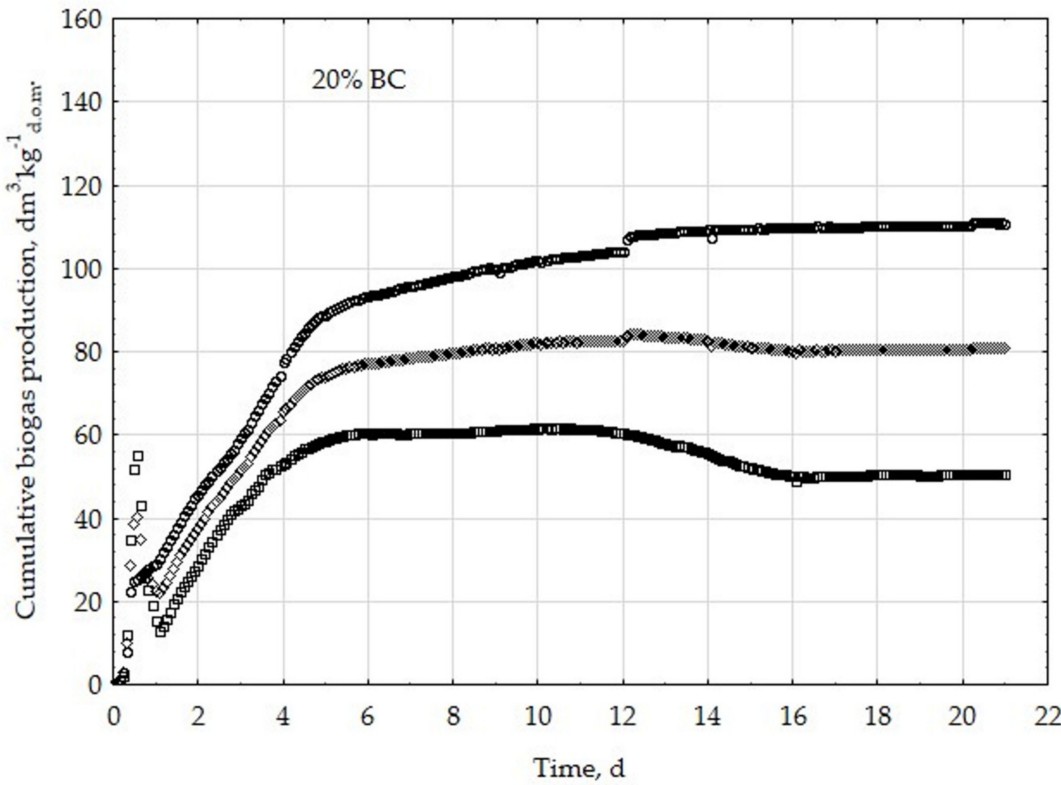

**Figure A7.** Biogas production from BSG: cumulative curves in variant with biochar addition at the level of 20%. The number of repetitions was three.

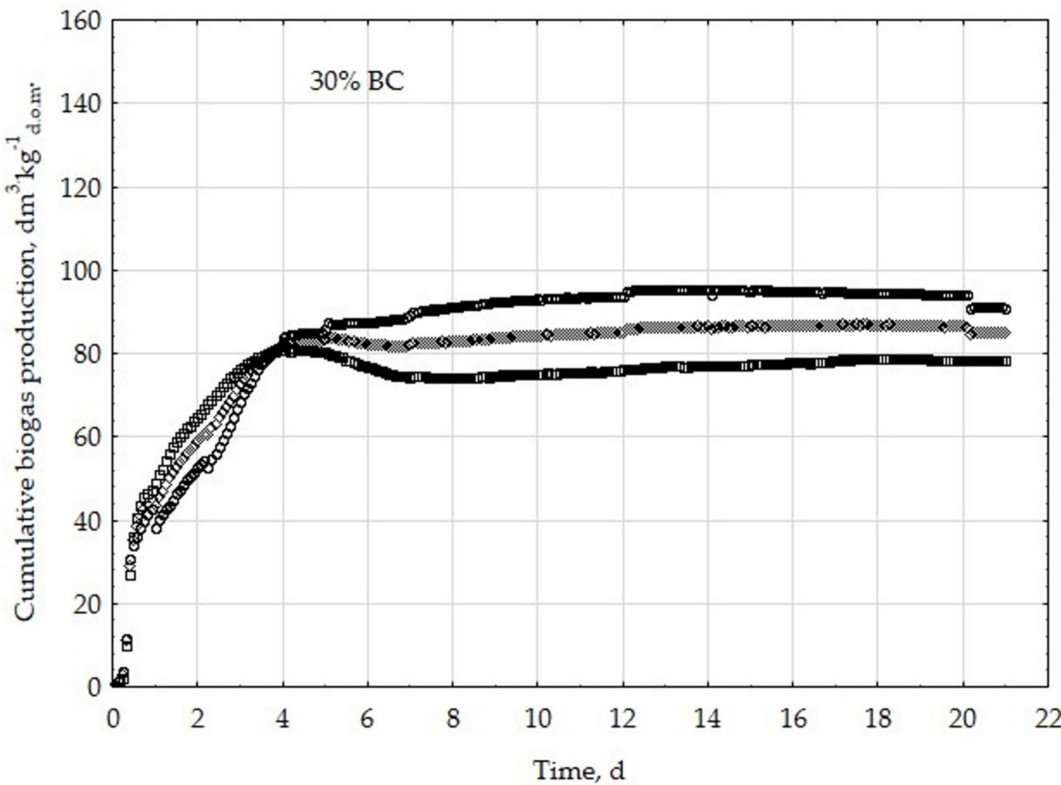

**Figure A8.** Biogas production from BSG: cumulative curves in variant with biochar addition at the level of 30%. The number of repetitions was three.

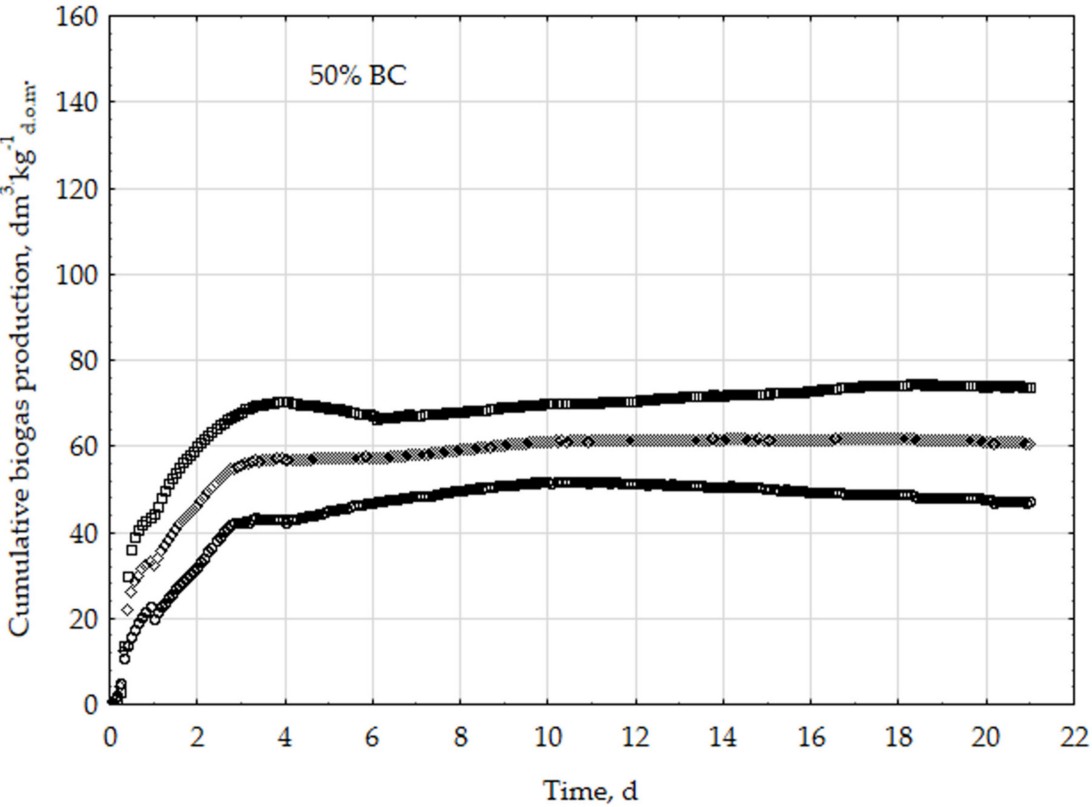

**Figure A9.** Biogas production from BSG: cumulative curves in variant with biochar addition at the level of 50%. The number of repetitions was three.

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
