# Peer review of "The Effect of Biochar Addition on the Biogas Production Kinetics from the Anaerobic Digestion of Brewers’ Spent Grain"

_energies, doi:10.3390/en12081518_

Round 1
Reviewer 1 Report
The paper titled “The effect of biochar addition on the biogas production kinetics from the anaerobic digestion of brewers’ spent grain” deals with an experimental study aimed at assessing the effect of biochar addition on biogas production from anaerobic digestion of brewers’ spent grain.
The English language is properly used and the paper is well written.
Nevertheless, the paper is not suitable for publishing in its current version and I invite the authors to revise the discussion of results and the conclusions taking into consideration the following points:
-the authors write that it was no dose effect in the addition of biochar but then they stress their conclusion saying that with a dosage of 5% they had an improvement of biogas production compared with no addition of biochar. This conclusion needs to be strengthened because considering figure 2, results from test added with 5% of biochar are almost totally included in the Standard deviation of the test with no addition of biochar;
-a decreasing trend in the biogas production can be noticed when the addition of biochar moves from 5% to 50%. The authors justify this negative trend saying that it could be due to neither (i) the “inhibiting effect of biochar on biogas production and microorganisms” and this sentence is in contradiction with the positive effect that authors found (5%) or (ii) “the sorption of biogas”. Have also the authors considered that the biochar was produced by heating the brewer’s spent grain at 300°C and at high temperature the organic matter could have lost the biodegradability because of the Maillard reaction (see doi:10.1016/j.jenvman.2014.07.042)? As all tests where conducted with the same amount of total VS (table 2) the occurrence of Maillard reaction could explain why where the dosage of biochar was higher the biogas production was lower.
To extend the introduction section I invite the authors to consider also this paper dealing with the abiotic processes that are involved in the degradation of organic matter in the anaerobic digestion: doi:10.3390/su6128348.
And to extend the discussion I invite authors to consider the following papers: doi: 10.3390/en9040247; doi: 10.1016/j.biortech.2018.12.068
Finally, at line 242, correct the sentence: “Furthermore, the is no…” in “Furthermore, there is no…”
Author Response
We submit the response to the reviewer in the attached file.

Reviewer 2 Report
This manuscript is focused on biogas production from brewers’ spent grain and its potential improvement by biochar addition. In the past years, facilitated methane production due to conductive materials (CM), including biochar, has been reported in the literature. Therefore, although this is a recent research field with relevant impact on bioenergy production from anaerobic digestion of wastes, the originality and novelty of this study is not high.
The M&M are nor clear – for example, I did not understand if, besides the inoculum+BSG+BC, something else was added to the bottles (e.g. culture medium, nutrients, alkalinity, reducing agent)? Was there any buffer? Moreover, the authors state that “Four research cycles were carried out; each cycle lasted 25 days (including lag-phase)” (lines 132-133) – What are these research cycles? Why these last 25 days if the GB21 assays last 21 days? In the substrates characterization, the composition in terms of proteins, sugars, Klason lignin, glucan and xylan content is also important and was not presented.
The presentation of the results is difficult to read and there is almost no discussion. Besides showing the results of B0, k and r, I think that the authors should present also the biogas production curves, so that we can see the raw data. Moreover, it would be useful to know, at least at the end of the experiments, what was the methane percentage in the biogas. The presented results have relatively high error bars (which is typical for AD of more heterogeneous wastes), but usually can be reduced by increasing the number of replicates. I was not sure if the data refers to triplicate assays, or triplicate assays performed 4 times (4 research cycles)? There is no information about the lag-phases, neither about other parameters such as pH, volatile fatty acids production, COD and solids reduction, that could complement the biogas production data and help to understand the differences in the biogas production.
Besides all that I have already mentioned, the major drawback of this work is that the results are not conclusive and thus the conclusions are not strongly supported by the data. In my opinion, this study requires further development, and therefore I consider that it is not adequate for publication in Energies.
Author Response

(The authors gave the same response as above.)

Reviewer 3 Report
The manuscript deals with an interesting study evaluating digestion of BSG and torrefaction of this same material for producing biochar.
The experimental set-up is well described and the manuscript in general is well written and easy to follow. However, some modification should be made to increase clarity of description and extra work should be performed, since the experimental results have not enough science impact
1. Abstract section should be significantly improved. At present is too long but essence of the research work is lost. I highly recommend to avoid excessive use of abbreviation when writing an abstract, since it is the first paragraph a new reader confronts, information presented should be easily found and understood. These characteristic increase the impact of the manuscript. In the abstract, there is an excessive use of the term BSG. Since biochar is not such a long word, then at least in abstract an introduction section keep it plain, even though it will be use later for describing experiments and samples.
2. L74-84. Pyrolysis process conditions (temperature and heating rate) affect biochar yield and characteristics, please add a small comment about this fact and justification for testing biochar produced from torrefaction (low temperature). Link with line 97-104
3. L 91, use the whole term dry matter, since it is not widely needed in the manuscript, no much sense of adding extra abbreviations
4. L94-96. Give a brief description of hydrothermal processes for char production and their main featuring against conventional pyrolysis
5. L105-114. Describe here just the aim of the manuscript. Although these are the working hypothesis, there must be placed previously in introduction section, so prior to describing methodology the reader must find the clear description of the aim of your work
6. Table 1 (also L127) should indicate properties of materials used in digestion tests, only BSG is the substrate while the table is including also those components needed for the whole set-up.
7. Table 1. Indicate measured properties are expressed in dry matter basis (in first row or table foot note), so d.m. can be eliminated in all rows. Make sure the same number of digits are used for the value and the deviation
8. L133-135, should be in the following section of experimental set-up, link with L144-146, eliminating duplicated information. The term BD is used but not previously defined, it seems a mistake for BC
9. L162. In this point, a description of nomenclature should be given
10. L167-168. Contains the same information of L164-165. Combine and delete duplicated information
11. L185. Use proper words, the index is expressed as volume of biogas (measured in dm3) per mass unit of dry organic matter (measured in kg), please correct, so the suitable volume and mass terms are included
12. Eq 4. This exponential model has been widely used for readily degradable substrates, please add references of a couple of studies where this model is used
13. L235. You may mean to have an effect “on” methane production, since the biological treatment of digestion is based on the microflora, not on the exclusive presence of biochar
14. Indicate the range of inoculum to substrate ratio of digestion test in Table 2
15. The experimental volume performed seems poor for a single manuscript. Results reported should be complemented considering a simplified energy balance at the condition of 5% addition. Authors are just reporting gas data from batch digestion test, which at the present state of the art seem insufficient. There is not even an evaluation of the final liquid effluent, resulting in an incomplete research. If this manuscript is to be considered for publication for a special issue, then further experimental work or energy analysis of the process should be provided
16. Gas evolution curves should be added, which I guess they are available, since otherwise, model constant would not have been calculated
17. Reduce conclusion section, there is no need of describing once again methodology, the research just goes to a simple conclusion, which is at 5% best result.
18. Please perform a wider search of journals, there are too many references from Elsevier journals, check also MDPI, of course, ACS, Springer and Taylor and Francis
Author Response

(The authors gave the same response as above.)

Round 2
Reviewer 2 Report
The authors made an effort to improve the manuscript, but the main methodological and scientific limitations remain in this version:
1) The authors state that they want to publish these results as a Communication, and as such the main experimental efforts were directed towards biogas production. However, in my opinion, the fact that methane was not measured and no other relevant data was presented (e.g. pH, volatile fatty acids) critically limits the scientific soundness of this work and makes any type of interpretation of the results impossible. Indeed, this “black box” approach only allows a yes or no answer to the questions “are B0/k/r higher in the presence of BC”, and the answer is NO in most cases.
2) A very important issue is that the authors performed statistical analysis of the results, but after they claim that a certain value is higher/lower than another even when the statistical analysis showed that the differences were not statistically significant. This is not correct from a scientific point a view, and cannot be published as it is.
3) In this study, there is no information about the biodegradability of the biochar tested, which is an important drawback because in each assay the increase of the biochar dose was accompanied by a decrease of the BSG dose. B0 values are expressed as dm3/kg d.o.m. and thus assays with higher biochar/lower BSG content can have a lower B0 if the used biochar is poorly biodegradable. This does not necessarily imply that the methane produced from the BSG was lower, and it can in fact be higher, i.e., it is possible that the biochar is increasing the biogas yield from the BSG (although this becomes hidden when the results are shown relatively to the organic matter of the mixture BSG+BC). Therefore, this information about BC biodegradability is very important. I strongly advise the authors to make a new set of experiments, similar to the ones described in the paper, amended with the same BC doses but without BSG. The results obtained should be included in the paper, and eventually the results already presented should be re-calculated/re-thought.
4) The authors refer this study as a proof of concept, but in my opinion this cannot be considered as such, but only a preliminary evaluation of the effect of co-digesting BSG with BC. Please change this on lines 25, 356 and 379.
In summary, for the manuscript to be considered adequate for publication, major revisions are still needed, and well as some minor changes. To make it easier to understand and modify, I will present them all together, going from the beginning until the end of the manuscript. I will underline the points that I consider more critical.
- Lines 18-21: I suggest to rewrite as follows: “In this research, for the first time, we test the feasibility of increasing biogas yield and rate from BSG digestion by adding BC, which was produced from BSG via torrefaction (low-temperature pyrolysis).”
- Line 24, “co-digested with nine doses of BC (0~50%)”: remove the term “nine doses”, it is enough to say that was co-digested with 0-50% BC. This percentage needs to be explained – only in the M&M we understand that it represents the mass ratio of BSG/BC, and event there I had the doubt if refers to wet or dry weight of the materials. Please clarify in both sections (abstract and M&M).
- Line 25, after “…n=3 trials.”: please add a sentence with the results obtained for the digestion of BSG (variant 0%) – B0, k and r.
- Line 26: explain what is d.o.m.
- Lines 28-34, from “The 5% BC dose…” until the end of the abstract: I suggest to rephrase as follows: “Due to the high variability observed between replicates, no significant differences could be detected between all the assays amended with CB and the variant 0% BC. However, a significant decrease of B0 in variants with the high biochar addition (20~50% BC) was observed in relation to 5% BC, suggesting that BC overdose inhibits biogas production from the BSG+BC mixture. The reaction rate constant (k) was not improved by BC, and the addition of 10% and 20% BC even decreased k relatively to the 0% variant. A significant decrease of k was also observed for the doses of 10%, 20% and 30% when comparing with the 5% BC assays.”
- I believe the manuscript has too many keywords. At least you may remove “beer” and DDGS.
- Lines 61 and 62: value of total biogas production potential of brewer's waste streams in the EU (12.6-39.7 x 109 MJ) – per year?
- Lines 67 and 68: the phrase starts by referring the BSG production in the EU, but after two values (EU and US) are presented. Please rectify this.
- Lines 87 and 88: “torrefaction” is repeated in the beginning and end of the phrase.
- Lines 137 and 138, “The torrefied biochar from BSG has a high concentration of C-containing compounds 137 that promote the growth of microorganisms needed for digestion.”: The authors have any experimental evidence that supports this (specifically for the studied BC)? If not, please remove this sentence.
- Please include the moisture content of the materials in Table 1.
- Line 159, “The main substrate for the fermentation process was BSG”: The authors have any experimental evidence that supports this? In the assays with high BC doses this may not be the case… Please provide evidence for this statement.
- Line 193, “Thus, the biogas production in D was subtracted from results obtained from other variants.”: please explain better what this means. The values from D were subtracted point by point? Or only the calculated B0, k and r? Please add also in supplementary, the methane production recorded in these assays (D, control).
- Line 196, “The initial dry matter content in each variant was 15%...”: this refers only to the contribution by BSG and BC, right? Please clarify in the paper.
- Line 197, “inoculum (D) to substrate (BSG+BC) ratio was 5.67.”: this is in VS? Please clarify in the paper.
- Table 2: please clarify if the values presented are expressed in wet weight or dry weight.
- Please revise and modify all the results and conclusions sections – values that are not statistically different should not be presented as such! This will represent a major change of the manuscript!
- If possible, add data from the biodegradability of the tested BC.
Author Response
We responded to the Reviewer's comments in the attached file.

Reviewer 3 Report
The manuscript have been corrected based on suggestion. However, the scientific content of the manuscript seems still low, if not additional research was to be reported as requested with this second round of corrections, at least an energy evaluation of the approach should be provided (a small energy evaluation of the combined process of digestion and addition of char derived from torrefaction. )
Author Response

(The authors gave the same response as above.)

Round 3
Reviewer 2 Report
The authors assume that BC was not biodegradable. In this case, BSG is digested in the presence of BC, and not co-digested. Please change this along the manuscript. Also in the M&M section, BC should not be presented as a substrate (line 147, Table 2) – please change.
Author Response
Our response to reviewer comment is in the attached file.
